# Real-Time Evaluation of Perception Uncertainty and Validity Verification of Autonomous Driving

**DOI:** 10.3390/s23052867

**Published:** 2023-03-06

**Authors:** Mingliang Yang, Kun Jiang, Junze Wen, Liang Peng, Yanding Yang, Hong Wang, Mengmeng Yang, Xinyu Jiao, Diange Yang

**Affiliations:** School of Vehicle and Mobility, Tsinghua University, Beijing 100084, China

**Keywords:** autonomous driving, uncertainty estimation, deep learning, perception uncertainty, object detection, spatial uncertainty, deep ensemble, prediction entropy

## Abstract

Deep neural network algorithms have achieved impressive performance in object detection. Real-time evaluation of perception uncertainty from deep neural network algorithms is indispensable for safe driving in autonomous vehicles. More research is required to determine how to assess the effectiveness and uncertainty of perception findings in real-time.This paper proposes a novel real-time evaluation method combining multi-source perception fusion and deep ensemble. The effectiveness of single-frame perception results is evaluated in real-time. Then, the spatial uncertainty of the detected objects and influencing factors are analyzed. Finally, the accuracy of spatial uncertainty is validated with the ground truth in the KITTI dataset. The research results show that the evaluation of perception effectiveness can reach 92% accuracy, and a positive correlation with the ground truth is found for both the uncertainty and the error. The spatial uncertainty is related to the distance and occlusion degree of detected objects.

## 1. Introduction

Perception, decision-making, and control are three core modules in autonomous driving(AD) [1,2]. Anomalies and uncertainty in perception directly impact how AD systems thoroughly comprehend the world and make driving decisions [3,4]. For AD systems to operate safely, the estimation of perceptual uncertainties must be done in real time and with accuracy [5]. Perception algorithms significantly influence the effectiveness of perception outcomes [6,7,8]. AD perception algorithms have undergone the development stages of the rule model algorithm, machine learning, and Deep Neural Network (DNN) algorithms [9].

Among these, the perception algorithm based on the rule model performs parameter optimization through feature extraction and manual modeling in a “top-down” manner, which has poor versatility and low efficiency [10,11]. DNN algorithms driven by data can handle situations that regular model algorithms cannot handle. Especially when the training data are sufficient, DNN algorithms show better comprehensiveness and accuracy of perception [12,13]. With the development of computer hardware computing power, storage, and other technologies, the proportion of DNN algorithms in autonomous driving systems is growing higher and higher. However, DNN algorithms have uncertainty and poor interpretability due to dataset uncertainty, training process uncertainty, and network internal structure factors [14,15,16].

As shown in Figure 1, perception imperfection can have a crucial influence on AD, especially when missed detection and large spatial uncertainties occur around the vehicle. When the current perception has a serious error, the failure should be detected in real-time for possible emergency response due to safety requirements.

Existing research focuses on the quantitative evaluation methods, influencing factors, trigger mechanisms, and processing methods of DNN uncertainties [7,12,17]. However, most of these references focus on uncertainty analysis with only one DNN; there are few references on mutual inspection based on multiple neural networks. It is difficult for a single neural network to avoid missed detection, which is a typical long-tail scenario in perception. Many approaches have emerged for false detection [18], perception anomalies [19], and perception risk assessment [5,20,21]. Most of these methods are judged based on the continuity of the perception data stream and the confidence index of the object output. These methods lack matching and verification with other data, meaning that the accuracy must be demonstrated. In addition, there are research methods for studying the perception uncertainty from the perspective of the safety of the intended functionality (SOTIF) [6,22,23]. These studies focus more on capturing uncertainty and researching the trigger mechanisms of uncertainty. The utilization of uncertain results can be applied in decision-making and the adjustment of the internal structure of the network [24,25,26]. Even if certain references extract uncertainty and process the uncertainty inside the network or through a decision-making algorithm, there is no quantitative demonstration of the correctness of the extracted uncertainty. In addition, many methods cannot be implemented in real-time.

A complete and accurate uncertainty assessment must face three core issues. How can we extract uncertainty completely to deal with long-tail scenarios under missed detection conditions in real time? How can the extracted uncertainty be verified without the ground truth of the dataset? What are the factors influencing neural network uncertainty?

Therefore, how to judge the effectiveness and uncertainty of perception results based on the matching of multi-source perception results in real-time needs further research. This work proposes a novel real-time evaluation method for AD perception. We select two kinds of objects detection DNN algorithms as our basic perception methods: Pointpillars, based on LiDAR [27], and SMOKE (Single-Stage Monocular 3D Object Detection via Keypoint Estimation camera), based on camera [28]. Then, we analyze the uncertainty results to evaluate the effectiveness of the perception results in real-time. Furthermore, quantitative analysis is carried out on the spatial uncertainty of the detected objects. Finally, the uncertainty results analyzed are verified based on the ground truth. In addition, the factors influencing DNN uncertainty are demonstrated quantitatively.

Our contributions can be summarized as follows:

(1) A real-time perception effectiveness estimation algorithm is proposed combining multi-source perception fusion and a deep learning ensemble. The model can judge the effectiveness of perception results and capture the spatial uncertainties of detected objects. This model can handle missed detections that are not intractable for a single network.

(2) While judging the perception effectiveness in real-time, the model can extract the spatial uncertainty of detected objects in real-time. Our study of the correlation of the ground truth with the uncertainty and error verifies the proposed model’s effectiveness.

(3) The perception effectiveness and spatial uncertainty obtained based on this model are verified on the KITTI dataset, demonstrating the correctness and accuracy of the model. The influencing factors of perception uncertainty are analyzed and verified.

The research in this paper can provide theoretical and practical guidance for real-time judgment of the effectiveness of AD perception results.

The remainder of the paper is structured as follows: Section 2 summarizes related works about perception uncertainties. Section 3 introduces the methodology used to evaluate uncertainty in this paper. Section 4 discusses the experimental results and demonstrates the relationship between the extracted uncertainties and the ground truth error. Finally, Section 5 summarizes our conclusions and describes possible future work.

## 2. Related Works

Significant studies have been devoted to capturing the perception uncertainty of DNN algorithms and to judging and monitoring perception anomalies.

Concept uncertainty, dataset uncertainty, network training, and migration uncertainty influence the uncertainty of perception based on DNN algorithms [29,30,31]. The Uncertainty Classification can be summarized into epistemic uncertainty and aleatoric uncertainty. Perception uncertainty theories include Bayesian theory, sampling-based methods, Gaussian theory, evidence theory, and propagation mechanisms [6,32,33]. For specific research methods, the Bayesian model can effectively capture uncertainty, although the computation cost is high.

Subsequently, Laplace Approximation (LA), Variational Inference (VI), and Markov Chain Monte Carlo (MCMC) have optimized the Bayesian model. For epistemic uncertainty, the Monte Carlo Dropout (MCD) [30] method and Deep Ensemble (DE) sampling [34] have been proposed. Compared with the MCD method, DE has higher calculation accuracy and meets real-time requirements.

Regarding uncertainty evaluation metrics, Di Feng [30] summarized the precision, recall, and F1-Score, along with the mean average precision, Shannon entropy, mutual information, calibration plot, error curve, etc. Zining Wang [35] proposed a new evaluation metric, Jaccard IoU (JIoU), that incorporates label uncertainty. Stefano Gasperini [36] provided separate uncertainties for each output signal: objectness, class, location, and size, and proposed a new metric to evaluate location and size uncertainty.

In terms of uncertainty influencing factors, Di Feng [37] proved that uncertainty is related to multiple factors, such as the detection distance, occlusion, softmax score, and orientation. Hujie Pan [38] found that the corner uncertainty distribution agrees with the point cloud distribution in the bounding box, meaning that the corner with denser observed points has lower uncertainty.

Evaluated perception uncertainty can be applied in the following aspects: improving perception performance, optimizing decision-making control algorithms, and providing warning and monitoring for AD systems. Gregory P. Meyer [13] estimated the uncertainty in detection by predicting a probability distribution over object bounding boxes, and proposed a method to improve the ability to learn the probability distribution by considering the potential noise in the ground-truth labeled data. Di Feng [37] leveraged heteroscedastic aleatoric uncertainty to improve its detection performance significantly. Q.M Rahman [5] studied run-time monitoring of machine learning for robotic perceptions based on perception uncertainty. In addition, from the perspective of SOTIF, Liang Peng [6,23] proposed a trigger mechanism for perception uncertainty and optimized the decision based on the uncertainty extracted.

In all, the mentioned references have greatly contributed to the evaluation methods and applications of perception uncertainty. However, the research object is usually one single detection network. In this situation, although the spatial uncertainty can be studied, it is difficult to deal with false detection and missed detection. This calls for mutual inspection of multiple perception algorithms. Moreover, perception failure needs to be detected in real-time to ensure safety. However, this is seldom studied in existing research. Therefore, it is important to judge the effectiveness of the current perception results. Finally, the analyzed perception uncertainty is a learning-based guess as to the relation of the error to the ground truth. Therefore, studying the relationship between real-time uncertainty evaluation, the error, and the ground truth is important. Thus, we have made an effort to contribute to these three problems, as stated in the Introduction.

## 3. Methodology

This section first introduces the theory of the DE method for uncertainty assessment, the clustering algorithm for multi-network results, and the principle of the matching algorithm for multi-source perception results. These three algorithms are the basis for the judgment of perception effectiveness and spatial uncertainty. Then, the perception effectiveness judgment algorithm and evaluation metrics for spatial uncertainty are discussed.

The logic flow between different algorithms is shown in Figure 2. PointPillars and SMOKE, shown in Figure 2, are two DNN algorithms for object detection, and represent two sources of perception. The DNN algorithm can detect objects using multiple neural networks after DE processing. The objects detected by different networks can be classified into different objects after being processed by the clustering algorithm. After the matching algorithm matches the objects from the two perception sources, the effectiveness of the current perception scene can be judged in real-time and combined with the remaining clustering results. Based on the clustering results, the spatial uncertainty of the detected objects can be solved directly.

The schematic sequence of the perception effectiveness and spatial uncertainty evaluation is shown in Figure 3. As shown in Figure 3, the uncertainty of detection results is primarily collected with Deep Ensemble. With multiple perception inputs, mutual inspection is carried out to analyze missed/false detection. Then, through a statistical study, the perception effectiveness is judged. If the result is judged to be effective, the spatial uncertainty of the outputs is estimated. Finally, with the ground truth of the objects from the dataset, the effectiveness judgment and the spatial uncertainty estimation result can be verified.

### 3.1. Deep Ensemble

The ensembles consist of two kinds of methods: randomization–based approaches and boosting-based approaches [34].

Bayesian neural networks provide a natural explanation for uncertainty estimation in deep learning [39]. The posterior distribution parameters in the Bayesian formula are related to the network parameters. Randomization–based sampling of the network parameters is required to simulate the output distribution. The Monte Carlo method can be used to sample the neural network parameters randomly, then the statistics can simulate this distribution [40]. The deep ensemble method based on randomization-based approaches is a simplified method. The deep ensemble can achieve random fine-tuning network parameters within a certain range. This pre-parameter sampling method does not increase the amount of calculation used in reasoning. Different sampling networks’ operation processes are the same, and can be calculated in parallel [41,42].

Therefore, the present study focuses on the randomization–based approach, as this approach is better suited for distributionand parallel computation. We use the DE approach to capture the real-time uncertainty of classification and regression outputs. The pseudo-code of the approach training procedure is summarized in Algorithm 1.
**Algorithm 1** Deep Ensemble**Input:**
Neural networks and number of networks ${\mathrm{Net}}_{1:N}$.
**Output:**
Associated classification probability $P_{1:N}$, location of detected objects $L_{1:N}$, rotation $R_{1:N}$, and dimension $D_{1:N}$.1:Construction and selection of neural networks.2:Parameter $W_{1:N}$ initialization of neural networks3:**for** n=1 to N **do**4:    Randomize the data loading mode for each network5:    Train different networks on the same dataset6:    Obtain classification and regression results7:**end for**8:**return** $P_{1:N}$, $L_{1:N}$, $R_{1:N}$, $D_{1:N}$.

In the training of neural networks, deep ensemble requires random initialization of the neural network parameters, along with randomized shuffling of data points, which is sufficient to obtain good training effects in practice.

### 3.2. Clustering Algorithm

The number and order of detected objects by different networks in the same frame scene are inconsistent; thus, the accurate matching of objects is an important guarantee of the accuracy and uncertainty of perception results. The objects merging strategy for sampling-based uncertainty used in this study is the basic sequential algorithmic scheme with intra-sample exclusivity (BSAS_excl) [17]. The approach training procedure is summarized in Algorithm 2.
**Algorithm 2** Basic sequential algorithmic scheme with intra-sample exclusivity**Input:** 
A set of predictions $P_{1:N}$.**Output:** 
A set of clusters $C_{1:N}$.1:Create a cluster for each for each object box in $C_{1}$.2:**for** i=2 to N **do**3:    Set excl_flag = $0_{n}$, n is the number of clusters4:    **for** box_j in $P_{i}$, for cluster_k in $C_{n}$ **do**5:        **if** affinity(box_j,cluster_k)≥θ and cluster_k=0 **then**6:            Put box_jincluster_k,excl_flag(k)=17:        **else**8:            Create a new cluster, n = n + 19:        **end if**10:    **end for**11:**end for**12:**return** C.

### 3.3. Object-Matching Algorithm

If there are multiple sources of perception, the perception results from different sources can be matched for further mutual inspection. Multi-source perception mutual inspection is an important method to improve perception performance. The object-matching algorithm is the basis of multi-source mutual inspection. This paper uses an algorithm based on triangle similarity [43,44] matching, which can effectively deal with the matching errors caused by object position errors.

The scheme flow of objects matching algorithm is shown in Figure 4. The steps of object matching can be summarized from step(1) to step(7)

(1)Set object sets $S_{A}$ and object sets $S_{B}$. Calculate the number of object sets $N_{A}$ and the number of object sets $N_{B}$.(2)Calculate the minimum value of elements in the object sets $m=min(N_{A},N_{B})$.(3)If m=1, use the K-Nearest Neighbor (KNN) method to calculate the distance of all of the points in $S_{A}$ to all of the points in $S_{B}$. The distance between point A and point B can be calculated in Equation (Equation 1).
(1)$$d_{AB}=\sqrt{{\left(x_{A}-x_{B}\right)}^{2}+{\left(y_{A}-y_{B}\right)}^{2}+{\left(z_{A}-z_{B}\right)}^{2}}$$
where $d_{AB}$ represents the 3D space distance between two points with set distance threshold $\tau_{d}$. If $d_{AB}<\tau_{d}$, this shows that the two points can be matched. Otherwise, the two points cannot be matched, indicating that the identified object may be a false negative or false positive.(4)If m=2, it is necessary to add a point at a distance to form a triangle, and both object sets need to add a point. The detection range is set to be 50 m. In order to achieve a better matching effect, the coordinates of the object points selected in this study can be (500,500) and (499,499). Points farther than 50 m away are added to form triangles for matching. These points do not belong to the objects within the perceived range and will not be output. It should be noted that the two points cannot be the same, but the distance between them cannot be too large. After adding points, triangle matching method can be used, which is shown below.(5)If m≥3, the triangle matching method can be used. The diagram of the triangle matching algorithm is shown in Figure 5.(6)The principle of the triangle matching method can be described from step (1) to step (7). Taking the object in the BEV perspective as an example, this paper only considers the index elements in x and z directions in the KITTI dataset camera coordinate system.(1)Numbering data to all of the points in $S_{A}$ and $S_{B}$.(2)Calculate the coordinate difference values $x_{disA}$, $z_{disA}$, $x_{disB}$, $z_{disB}$ between the maximum and minimum values in *x* and *z* directions in object sets $S_{A}$ and object sets $S_{B}$ by using Equations (Equation 2) and (Equation 3).
(2)$$x_{dis}=x_{max}-x_{min},\forall x\in S_{A},S_{B}$$

(3)
$$z_{dis}=z_{max}-z_{min},\forall z\in S_{A},S_{B}$$

(3)Sort the points in object sets $S_{A}$ and object sets $S_{B}$. Calculate the maximum value of the coordinate difference values. If the maximum value is in the x direction, the object sets $S_{A}$ and object sets $S_{B}$ are sorted by the value of x. Otherwise, object sets $S_{A}$ and object sets $S_{B}$ are sorted by the value of z.(4)Randomly select three points to form a triangle in object sets $S_{A}$ and object sets $S_{B}$, and return the index of the point location.(5)The formed triangles in object sets $S_{A}$ and object sets $S_{B}$ are normalized. The side length of each triangle is divided by the shortest side length. This setting can ensure the setting of a uniform threshold for successful matching.(6)Calculate the error sum of the edges and points for each triangle in object sets $S_{A}$ with all triangles in object sets $S_{B}$. Take any two points in a triangle as an example. As shown in Figure 5, Point $A_{1}$ and point $A_{2}$ are corresponding points. The error of point A can be expressed in Equation (Equation 4).
(4)$$P_{Aerror}=\left|x_{A_{1}}-x_{A_{2}}\right|+\left|z_{A_{1}}-z_{A_{2}}\right|,\forall A_{1}\in S_{A},\forall A_{2}\in S_{B}$$The error of edge AB can be expressed in Equation (Equation 5).
(5)$$d_{ABerror}=\left|d_{A_{1}B_{1}}-d_{A_{2}B_{2}}\right|,\forall A_{1},B_{1}\in S_{A},\forall A_{2},B_{2}\in S_{B}$$The total error (TE) of two triangles can be expressed in Equation (Equation 6).
(6)$$T_{error}=d_{ABerror}+d_{ACerror}+d_{BCerror}+P_{Aerror}+P_{Berror}+P_{Cerror}$$(7)Calculate the minimum value $T_{error}$ of all trianglestriangle. If $T_{error}$ is less than the triangle error threshold $\tau_{T}$, the two triangles are matched, and the corresponding points of the triangle sorted by (3) are the matching objects.(7)Judge the remaining points, and repeat steps (3), (4), and (5) until all points are matched.

### 3.4. Perception Effectiveness Judgment

Each perception source will have missed and false detection; thus, it is necessary to judge the effectiveness of the current frame perception information in real time. If the perception results are invalid, it is necessary to make an early warning, or perception switching can be performed directly. If the perception results are valid, the spatial uncertainty of the detected objects is calculated to carry out further trajectory planning and the construction of the drivable space.

Therefore, here comes three key questions: (1) What are the criteria for judging the effectiveness of perception results? (2) How can we judge the effectiveness of current perception results in real time? (3) How can we verify the judgment results?

First of all, for the real-time perception effectiveness judgment, the precision, recall, and F1-score of the perception results of the current frame are applied as the criteria. The judgment threshold is based on the statistical results of the KITTI full data set. Second, for the method of perception effectiveness judgment, this paper adopts a fusion model based on multi-source perception and deep ensemble to judge the effectiveness of current perception results in real time. Third, to verify the judgment results, this paper matches the perception results with the ground truth of the dataset. And calculate the precision, recall, and F1 score of each frame information to judge the effectiveness of the perception results. Compare the validity results of the real-time judgment with the validity of the ground truth matching results to verify the validity of the judgment results. Finally, calculate the anomaly diagnosis rate.

The flowchart of perception effectiveness judgment is shown in Figure 6. The DE method is used to study the uncertainty of object occupation. The normal detection (True Positive, TP), missed detection (False Negative, FN), and virtual detection(false positive, FP) of the perception results are studied. If there is only one perception source, it can evaluate the object occupation uncertainty according to the number of networks $N_{net}$ and the average confidence $p_{m}$. If there are multiple perception sources, this paper uses the method combining deep ensemble and multi-source perception mutual acceptance to research occupancy uncertainty.

First, the steps of uncertainty research for one perception source are as follows.

(1)The perception results are processed in the DE method. After clustering and statistics, the number of networks detecting the object $N_{net}$ and the average confidence level $p_{m}$ of the detected object are calculated in Equation (Equation 7), respectively.
(7)$$p_{m}=p\left(y=c|D\right)=\frac{1}{N_{net}}\sum_{i=1}^{T}p\left(y=c|x,W_{t}\right)$$
where *c* represents the classification of detected objects. *D* represents the dataset to evaluate the detected objects and $W_{t}$ represents the weights of the network.(2)Set detection network number threshold $\tau_{N}$ and average confidence $\tau_{P_{1}}$ and $\tau_{P_{2}}$.(3)Judge the uncertainty by using Equation (Equation 8).
(8)$$f\left(N_{net},p_{m}\right)=\left\{\begin{array}{ccc} TP & , & ifN_{net}\geq \tau_{N}andp_{m}\geq \tau_{P_{1}}\\ FN & , & ifN_{net}<\tau_{N}andp_{m}\geq \tau_{P_{1}}\\ FP & , & ifN_{net}\geq \tau_{N}andp_{m}<\tau_{P_{2}} \end{array}\right.$$

If there are multiple sources of perception, the matching of different perception results is carried out first. Then the uncertainty research is carried out by using the DE method.

(1)First, the triangle matching method is used for object matching of multi-source perception results. Objects that are successfully matched are considered to be real objects.(2)For the results that cannot be matched in each perception source, the DE method is used for judgment.(3)The final processing results are fused.

In addition, for the uncertainty of classification and occupation of each detected object, the prediction entropy $E_{pe}$ is selected in this study as the uncertainty evaluation index, which is calculated in Equation (Equation 9). During the calculation of prediction entropy, the confidence levels of different networks are averaged, and then the prediction entropy is calculated
(9)$$E_{pe}=-\sum_{p_{m}}^{C}\left(p_{m}logp_{m}+\left(1-p_{m}\right)log\left(1-p_{m}\right)\right)$$

Because there is missed detection in the calculation process, it is necessary to punish the object detected to calculate the detected object’s uncertainty more objectively, which is calculated in Equation (Equation 10).
(10)$$E_{pe}^{*}=E_{pe}\left(1+p\left(T-N_{net}\right)\right)$$
where *p* represents the penalty coefficient.

### 3.5. Spatial Uncertainty

The spatial information in the regression of the neural network algorithm includes the horizontal position (*x*), vertical position (*z*), and vertical position (*y*) of the detected object, the length (*l*), width (*w*), and height (*h*) of the detected object, and the orientation (*r*) of the detected object. The corresponding uncertainty evaluation indicators can be represented by variance and total variance (TV).

The variance of each indicator can be expressed in Equation (Equation 11)
(11)$$var_{x}=\frac{1}{N_{net}}\sum_{1}^{N_{net}}{\left(x_{i}-\frac{1}{N_{net}}\sum_{1}^{N_{net}}x_{i}\right)}^{2}$$

Similarly, the variance of other indicators is calculated using the same equation. Each object has a location and dimension, and the TV of location and dimension can be calculated in Equations (Equation 12) and (Equation 13).
(12)$$TV_{loc}=var_{x}+var_{y}+var_{z}$$
(13)$$TV_{dim}=var_{l}+var_{w}+var_{h}$$

## 4. Experimental Results

In this section, experiments are carried out in the KITTI dataset to validate the proposed methods. First, we introduce the experiment settings, including the detection networks and the implementation details. Then, the quantitative results are introduced.

### 4.1. Experiment Settings

#### 4.1.1. PointPillars 3D Objects Detection Network

The Network Architecture of PointPillars is shown in Figure 7. The main components of the network are a Pillar Feature Network, Backbone and SSD Detection Head. The point clouds are converted to a pillar index tensor and stacked pillar. The encoder uses the stacked pillars to learn a set of features that can be scattered back to a 2D pseudo-image for a convolutional neural network(CNN). The features from the backbone are used by the detection head to predict 3D bounding boxes for objects. According to 3D bounding boxes, detection can be matched to evaluate the uncertainty of classification, regression, and a plurality of outputs.

The training implementation platform was CUDA11.6, CUDNN 8.8, and Pytorch1.10.1. We trained the network with a batch size of 2 on one Geforce TITAN V GPU for five epochs. The learning rate was set at 2 × $10^{-4}$ and drops at 15. During testing, we used the top 100 detected 3D projected points and filtered them with a threshold of 0.5. No data augmentation methods or NMS (Non-Maximum Suppression) were used in the test procedure.

#### 4.1.2. SMOKE 3D Objects Detection Network

The Network Architecture of SMOKE is shown in Figure 8. SMOKE predicts the three-dimensional bounding box of each detection object by combining the estimation of a single key point with the three-dimensional regression variables. SMOKE uses a high-level integrated network DLA-34 as the backbone. The high level is replaced by a deformable convolutional network (DCN). SMOKE proposes a multi-step separation method to construct a three-dimensional bounding box, which greatly improves the training convergence and detection accuracy. In the 3D detection network, the target of SMOKE is represented by a key point. With camera parameters, projecting the key point can completely restore the 3D position.

The training implementation platform was CUDA10.0, CUDNN 7.5, and Pytorch1.1. The original image resolution was padded to 1280 × 384. We trained the network with a batch size of 32 on four Geforce TITAN X GPUs for 60 epochs. The learning rate was set at 2.5 × $10^{-4}$ and dropped by a factor of 10 at 25 and 40 epochs. During testing, we used the top 100 detected 3D projected points and filtered them with a threshold of 0.25. No data augmentation methods or NMS (Non-Maximum Suppression) were used in the test procedure.

#### 4.1.3. Implementation Details

The experiment used the official KITTI object detection benchmark dataset, including samples pointing to clouds and images. The experiment simultaneously trained the lidar point cloud and compared it with the fusion method using lidar and image. The dataset sample included 7481 training samples and 7518 test samples; the experimental study divided the training set into 3712 training samples and 3769 validation samples. The classification results include cars, pedestrians and cyclists. PointPillars used a DNN network for cars and a DNN for pedestrians, and cyclists during training. In validation, the output results included three categories: car, bicycle, and pedestrian.

The perception results from PointPllars are based on lidar and those from SMOKE are based on a camera. The two DNN algorithms are three-dimensional object detection networks, and both are trained and evaluated on the KITTI dataset, meaning that that the results can be compared and matched. In DE parameter settings, both neural network parameters adapt kai-ming uniform distribution to achieve the initialization of parameters. The data loading model adapts the shuffle format. The total number of networks of PointPillars and SMOKE is set to 5, respectively.

In the parameter setting of the clustering algorithm, the three-dimensional space intersection over union (IOU) is set to 0.1 to better achieve the result matching. In the triangle matching algorithm, the KNN method and the triangle error threshold are set to 15. The above parameters are manually adjusted and optimized according to the effect of the actual experiment.

In the perception effectiveness judgment algorithm, there are key parameter thresholds such as the number of object recognition networks, object mean score, precision, recall, and F1-score of the current frame results. In this study, we counted and calculated the precision, recall, and F1-score of the objects detected in 3769 frames matching the KITTI ground truth by using PointPillars and SMOKE after DE, as shown in Table 1. In addition, we counted and calculated the number of object recognition networks and the average object score. We counted the number of detected networks and the average object score of the detected objects in 3769 frames using PointPillars and SMOKE after DE, as shown in Table 2.

Based on the above statistical results, this paper sets the number of detected networks and the average object score of PointPillars are 4 and 0.6, respectively. The number of detected networks and the average object score of SMOKE are 2 and 0.4, respectively. The thresholds that precision, recall and F1 scores, are set 76.5%, 47.1% and 0.583, respectively.

### 4.2. Results

This section quantitatively demonstrates the perception effectiveness, focusing on three typical cases. Then, the spatial uncertainty of the detected objects is extracted and the relationship between uncertainty and ground-truth error is demonstrated. Finally, this section verifies the relationship between perception uncertainty, object distance, and occlusion degree.

#### 4.2.1. Perception Effectiveness Judgement

This section counts and displays the judgment results of perception effectiveness from the macroscopic and microscopic perspectives. This section first counts the effectiveness results of 1000 frames on the KITTI dataset, including the results of correct judgment (including the direct judgment of no objects as invalid perception) and the results of wrong judgment.

The specific statistical results are shown in Table 3.

The correct results judged include three situations. (1) The current frame scene is judged valid, and the result is also valid after matching the ground truth. (2) The current frame scene is judged invalid, and the result is invalid after matching the ground truth. (3) The perception has no output result, which is judged invalid. However, the truth result shows there are objects, indicating that the judgment of perception effectiveness is correct. Based on the above analysis, the effectiveness judgment and verification of 1000 frames of the KITTI evaluation data set are carried out. The statistical results show that 920 frames are correct, which contains the perception results without outputs, and the judgment of 80 frames is wrong. Therefore, it can be concluded that the accuracy rate of perception effectiveness judgment can reach 92%. The statistical results of large samples in the data set can reflect the method’s effectiveness.

In addition, this paper selects the perception results that are judged to be valid and judged to be invalid. The perception effectiveness judgments are correct.

Figure 9 shows the results when the perception effectiveness judgments are correct. Among these results, the perception outputs are judged to be valid results, which are valid after matching and verifying with the ground truth of the KITTI dataset. Hence, the judgment of perception effectiveness is correct. From a microscopic point of view, this paper selects the 25th frame, the 542nd frame, the 634th frame, and the 1932nd frame in the KITTI dataset. Among them, the green boxes represent the ground truth, and the blue boxes represent the results of the fusion model matching of PointPillars and SMOKE, which are considered real perception results. The red boxes are the perception results that PointPillars considered detected objects after the DE method based on the number of detected networks and the average object score. The yellow boxes are the perception results that SMOKE considered detected objects after the DE method based on the number of detected networks and the average object score. The red FN and FP represent the missed detection and false detection of PointPillars after the DE method based on the number of detected networks and the average object score, respectively. The blue FN and FP represent the missed detection and false detection of SMOKE after the DE method based on the number of detected networks and the average object score, respectively. The figure’s background is the original point cloud information projection based on lidar under the grid map, where white represents the space occupied, and black represents the unoccupied space. The depth of the color represents the probability of being occupied.

Figure 9 shows that the bounding boxes between the perception results and the ground truth results do not match exactly. This is mainly due to the uncertainty of the perception algorithm. Therefore, there are spatial errors in the perception results which generally do not affect AD safety. In the perception results of the 25th frame, there is no missed detection. Outside the range of the ground truth, a few objects are detected. These objects basically match the occupancy information of the original point cloud. Hence, they must be considered in the subsequent drivable space construction and decision-making process, which is more helpful for AD safety. In the 542nd and 634th perception results, although there is a missed detection at the farthest point, the update frequency of the sensor is relatively fast, and the missed detection in the far distance will not cause driving safety of AD. In the perception result of the 1932nd frame, SMOKE detects a missed detection, which is consistent with the occupancy information of the original point cloud in the grid map. This indicates that the dataset may have errors in data labeling in certain cases. The perception result of this frame is more conservative and accurate, which can guarantee the safety of AD.

Figure 10 shows the results that the perception effectiveness judgments are correct.The perception outputs are judged to be invalid, and are invalid after matching and verifying with the ground truth of the dataset, meaning that the judgment of perception validity was correct. From a microscopic point of view, this paper selects the first frame, the 702nd frame, the 949th frame, and the 979th frame in the KITTI dataset. The information represented by different colors and boxes in the figure is consistent with that in Figure 9.

Among these perception results, there are cases of missed detection and false detection that are inaccurate, and there is a significant gap with the true value. If there are missed detections in the perception results, this directly impacts the safety of AD. Timely evaluation and identification of these perception failure scenarios can optimize the decision-making strategy and improve the safety of AD.

Figure 11 shows the situation that the perception algorithm has no output. This paper selected the 81st and 260th frames in KIITTI. The green box represents the ground truth, where the perception algorithm has no perception output result; thus, this paper concludes that these frames’ perception results are invalid. The perception effectiveness is judged to be correct.

The results above show that the perception effectiveness can be evaluated in real time based on multi-source mutual inspection and the proposed DE fusion algorithm. If the current perception result is valid, the spatial uncertainty of the detected objects can further support accurate decision-making. If the perception result is invalid, it can be handled by perception switching, manual takeover, or emergency stop to ensure the safety of AD.

#### 4.2.2. Spatial Uncertainty

The spatial uncertainty mainly includes the location of the perception results. The mean and variance of these perception results based on DE are calculated, respectively. In addition, by comparing the perception results with the ground truth of the dataset, the calculation error can also represent the uncertainty of the perception results.

This paper selects a frame of perception results to show spatial uncertainty. Figure 12 shows the ground truth of the dataset, perception results, uncertainty of perception results, and errors between perception results and ground truth. The green boxes indicate the ground truth of the dataset, and the blue boxes indicate the perception results. Red boxes represent that the standard deviation perception result is considered based on the DE method, which indicates the uncertainty. Yellow represents the perception result considering the errors between the perception results and ground truth. The comparison in the figure can reflect the size of the perception error. These perception errors and other uncertainty can provide a reference for decision-making and ensure the safety of AD.

The results of spatial uncertainty show that the detected objects of DNN algorithm have uncertainty in location. The location uncertainty of different objects is different, and this gap is very large in some cases. Among them, the influencing factors of uncertainty are a problem worth studying, and represent a meaningful topic to further study the accuracy of spatial uncertainty.

#### 4.2.3. Validation of Perception Uncertainty

In order to verify the accuracy of the spatial uncertainty of the detected objects, this paper calculates the error between the perception results and ground truth. And the paper demonstrates the correlation between uncertainty and error results. This paper researches the correlation of the position and orientation uncertainty of objects detected based on PointPillars and SMOKE, respectively.

Figure 13 shows the correlation of uncertainty extracted and the error between perception results and ground truth in PointPillars. The correlation coefficients of the horizontal, longitudinal, vertical, and orientation of the ego vehicle using PointPillars in the entire KITTI dataset are 0.317, 0.299, 0.168, and 0.657, respectively.

Figure 14 shows the correlation of the extracted uncertainty and the error between perception results and ground truth in SMOKE. The correlation coefficients of Horizontal, longitudinal, vertical, and orientation of the ego vehicle using SMOKE in the entire KITTI dataset are 0.184, 0.159, 0.058 and 0.569, respectively.

The results indicate that there is a positive correlation between uncertainty and errors between perception results and ground truth based on DNN algorithms, which shows that the method of extracting uncertainty is scientific and accurate.

#### 4.2.4. Influencing Factors of Uncertainty

This paper studies the relationship of the distance and occlusion factors with the perception uncertainty. This paper analyzes the perception results within 50 m. In studying the distance factor, the distance is divided into 20 categories with a step size of 2.5 m. Objects within the same distance interval are counted and classified into one class. Then, we calculate the average uncertainty of these objects. The occlusion factors are divided into 0,1,2,3 levels, meaning that the statistical results of objects detected uncertainty in the four levels are counted. Then the relationship between occlusion and uncertainty factors is studied.

The red line in Figure 15 shows the trend between the distance factor and prediction entropy, TV, and TE of location. The blue line is the scatter plot line between these factors. From a quantitative perspective, the Pearson coefficient between the prediction entropy, TE of PointPillars, and distance are 0.584 and 0.107, respectively. the Pearson coefficient between prediction entropy, TV of SMOKE, and distance are 0.899 and 0.348, respectively. It should be noted that within a distance of 10 m, the size of the uncertainty shows a downward trend in a local area, which is mainly related to the sensors’ installation position and the sensor characteristics. If the sensor is installed on the ego vehicle roof, the observation effect of the objects detected will not be good, resulting in increased uncertainty.At the same time, in the ultra-short range, the sensor resolution decreases, which increases the perception uncertainty. Generally speaking, there is a positive correlation between perception uncertainty and distance beyond a certain distance. However, within the ultra-short range of the ego vehicle the poor detection effect of objects may cause higher perception uncertainty.

The red line in Figure 16 shows the trend between the occlusion factor and the prediction entropy, TV, and TE of location. The blue line is the scatter plot line between these factors. The Pearson coefficients between prediction entropy, TV, and TE of Pointpillars are 0.665, 0.505, and 0.987, respectively. In general, the higher the degree of occlusion, the higher the uncertainty of objects detected. However, the prediction entropy and TV in this study show a downward trend when the occlusion degree is 3. This is related to the data distribution of the KITTI dataset. There are very few sample data with an occlusion degree of 3, resulting in statistical result errors. However, there is a positive correlation between the perception uncertainty and the degree of occlusion in the sample.

In conclusion, the experimental results proved:(1)The proposed real-time judgment on perception effectiveness has a high accuracy of 92%.(2)The estimation of spatial uncertainty based on DE is positively correlated to the ground truth error.

In addition, the influencing factors of perception uncertainty are discussed.

## 5. Conclusions

This paper proposes a fusion model based on multi-source perception and Deep Ensemble to judge the effectiveness of perceptual results in each frame and evaluate the spatial uncertainty of the objects detected simultaneously. Based on the KITTI dataset, the research results of this paper show that the accuracy of judging the effectiveness of perception based on the multi-source perception inspection and Deep Ensemble fusion model can reach 92%. In addition, a positive correlation is found between perception spatial uncertainty and error between perception results and ground truth. The results grant the uncertainty evaluation a physical meaning anchored to the objective error. In addition, this study found that perception uncertainty is related to the distance and the degree of occlusion of the detected objects.

Compared with the previous research, the research in this paper can effectively and real-time deal with the missed detection in the long tail scenario and judge the perception effectiveness, which is very important for the safe operation of autonomous driving. The research in this paper can effectively improve the accuracy of perception and further improve the safety of autonomous driving. Suppose a perception failure is found in real-time. In such a case, it can be replaced with other perception sources to realize real-time judgment and switching of autonomous driving perception sources and ensure the safety of autonomous driving. Although we have achieved good results, we only studied the uncertainty of perception occupancy, which needs to be verified on real vehicles. In the future, we hope to further study the uncertainty of semantics and motion of perception results to improve AD’s perception performance and safety. The model proposed in this article is incredibly important for assessing perception uncertainty in real-time, and could benefit autonomous driving safety even more.

## Figures and Tables

**Figure 1 sensors-23-02867-f001:**
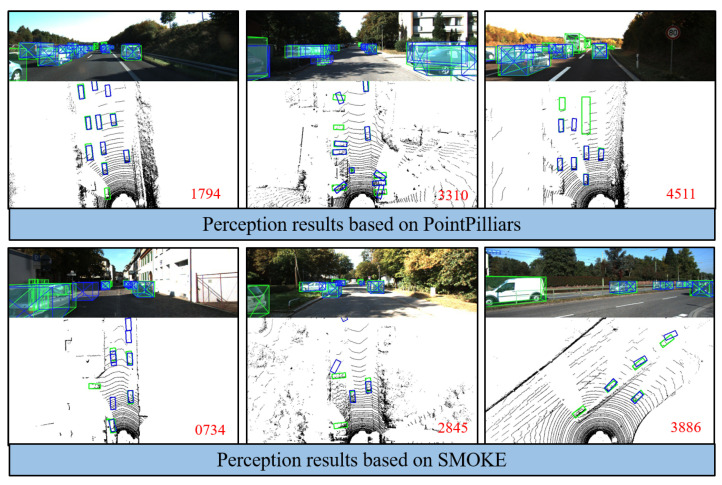
Comparison graph between ground truth and perception results under the KITTI dataset based on DNN algorithms. There is uncertainty in perception results, such as missed detection, false detection, position errors, and orientation errors. Green boxes represent the ground truth of data labels and blue boxes donate the perception results of DNN algorithms based on lidar and camera. The red numbers represent the order of the dataset frames.

**Figure 2 sensors-23-02867-f002:**
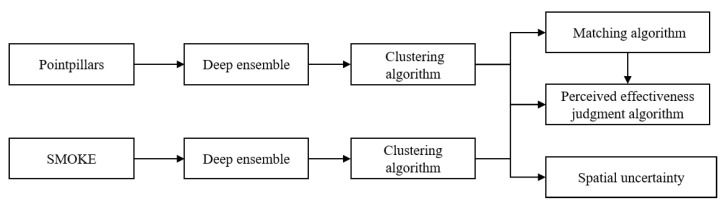
The logic flow between different algorithms.

**Figure 3 sensors-23-02867-f003:**
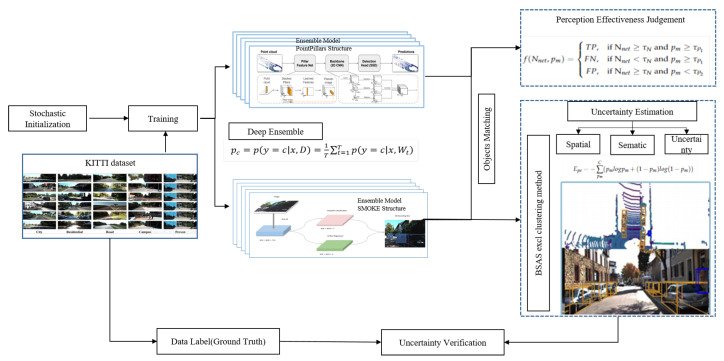
The Schematic sequence of the perception effectiveness judgment and uncertainty evaluation.

**Figure 4 sensors-23-02867-f004:**
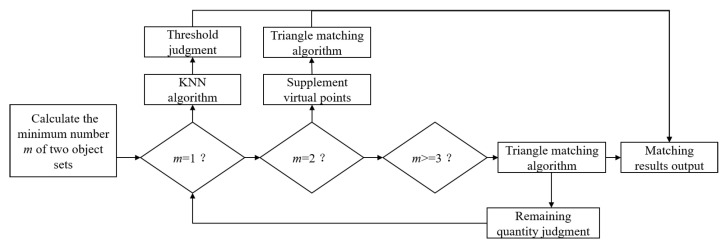
Scheme flow of objects matching algorithm.

**Figure 5 sensors-23-02867-f005:**
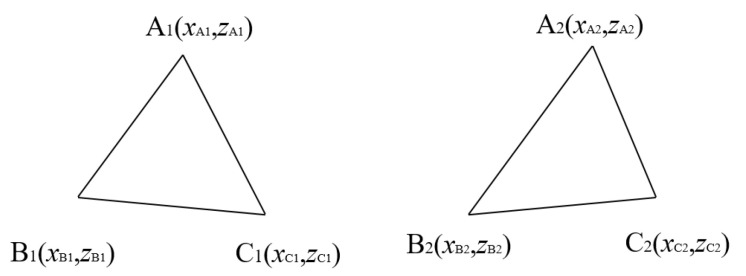
Objects matching algorithm:Diagram of triangle matching.

**Figure 6 sensors-23-02867-f006:**
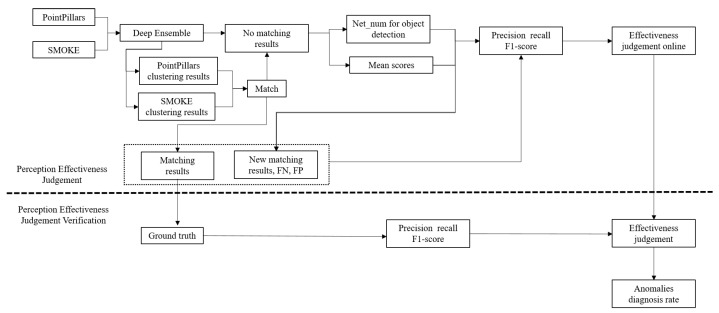
Scheme flow of the perception effectiveness judgment.

**Figure 7 sensors-23-02867-f007:**
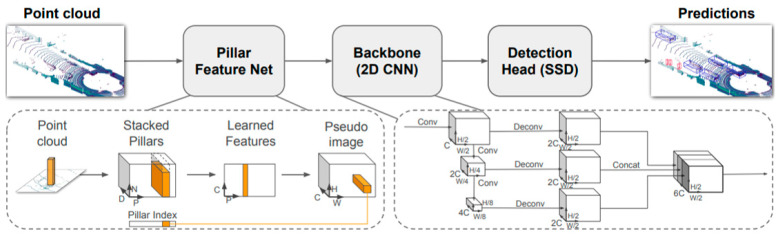
Scheme flow of PointPillars [27].

**Figure 8 sensors-23-02867-f008:**
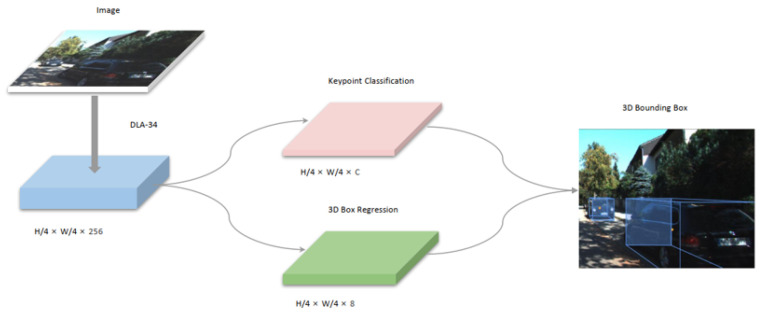
Scheme flow of SMOKE [28].

**Figure 9 sensors-23-02867-f009:**
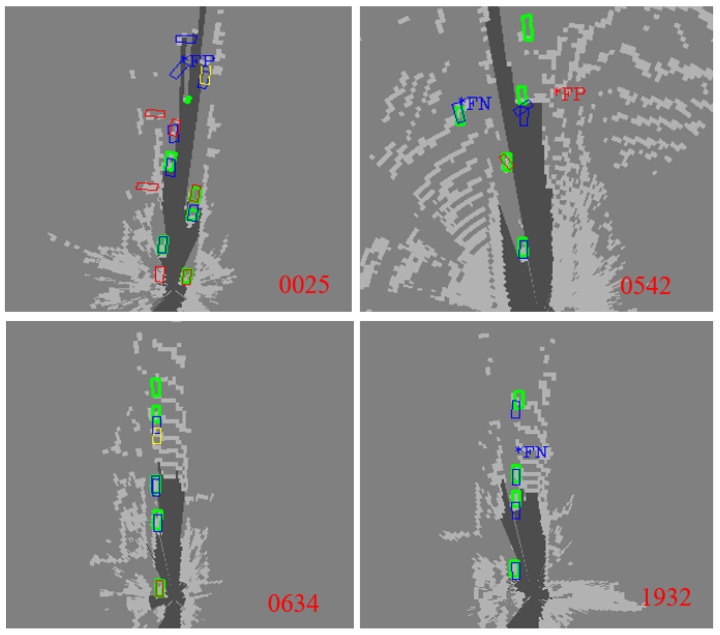
Judgment results of perception effectiveness: the judgment is correct. The judgment and verification are valid after matching and verifying with the ground truth.

**Figure 10 sensors-23-02867-f010:**
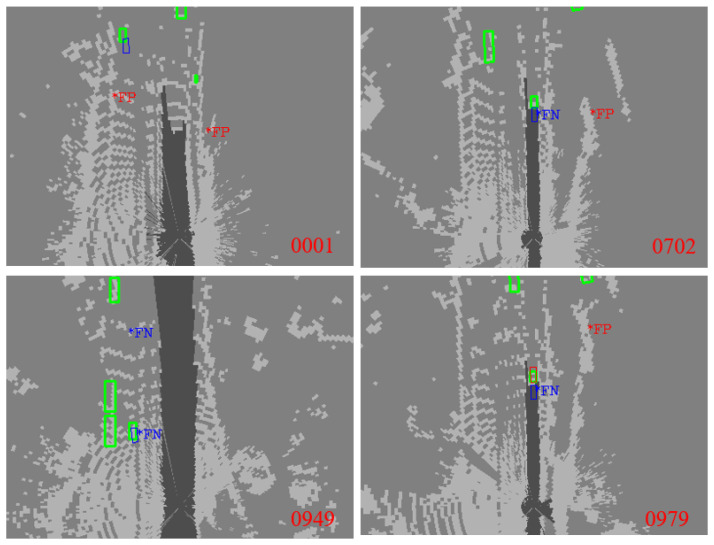
Judgment results of perception effectiveness: the judgment is correct. The judgment and verification are invalid after matching and verifying with the ground truth.

**Figure 11 sensors-23-02867-f011:**
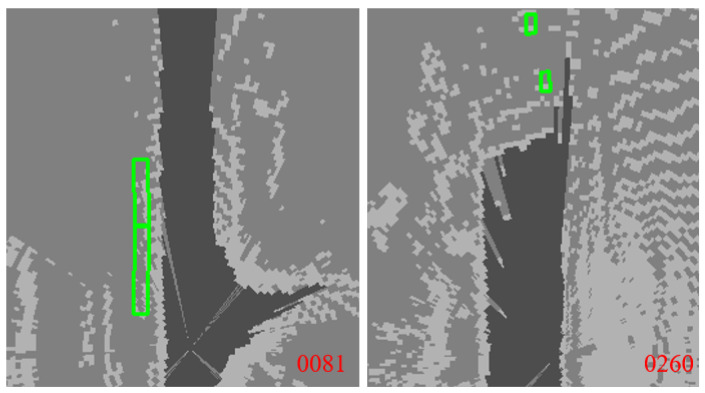
Judgment results of perception effectiveness: the judgment is correct. The judgment and verification are invalid after matching and verifying with the ground truth.

**Figure 12 sensors-23-02867-f012:**
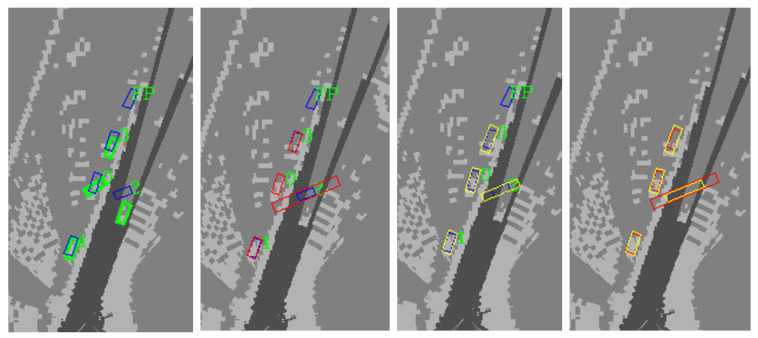
Spatial uncertainty based on deep ensemble.

**Figure 13 sensors-23-02867-f013:**
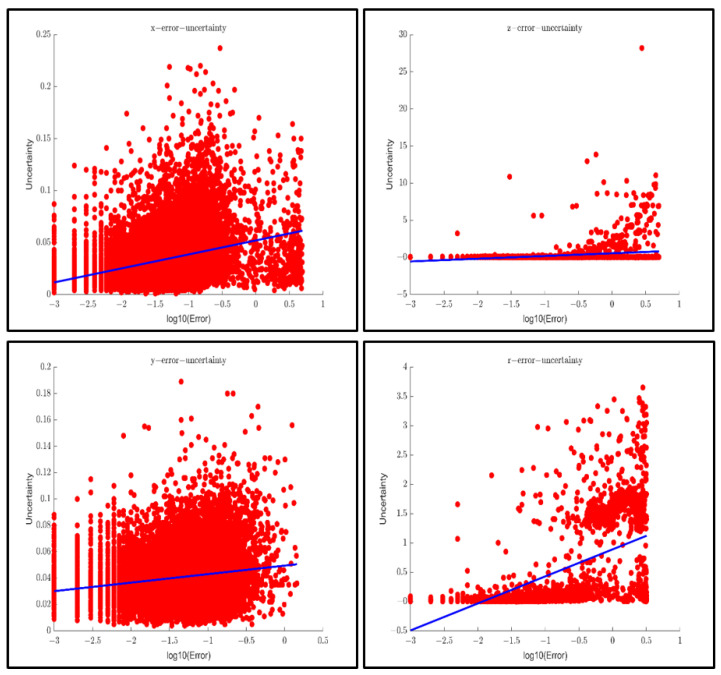
Correlation research between uncertainty and error of Pointpillars (3769 frames) In order: the horizontal direction, longitudinal direction, vertical direction, and orientation of the car.

**Figure 14 sensors-23-02867-f014:**
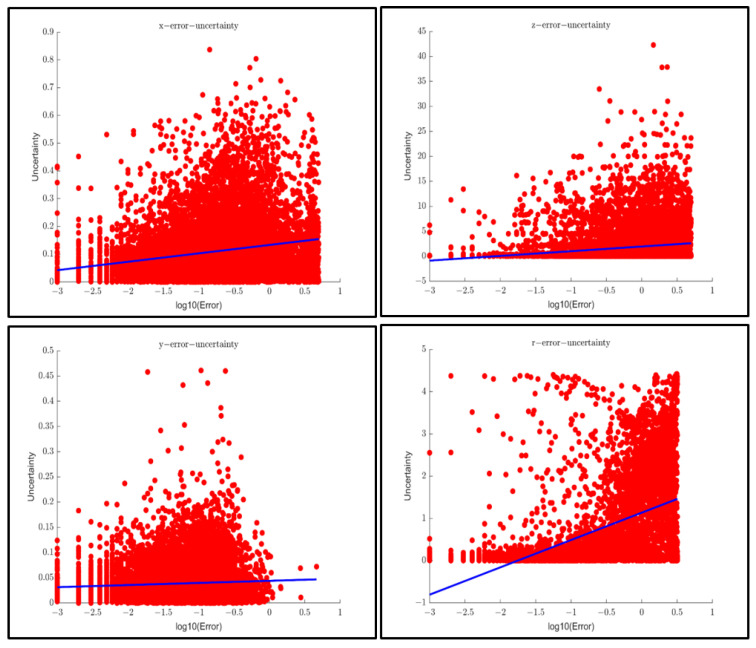
Correlation research between uncertainty and error of SMOKE (3769 frames) In order: the horizontal direction, longitudinal direction, vertical direction, and orientation of the car.

**Figure 15 sensors-23-02867-f015:**
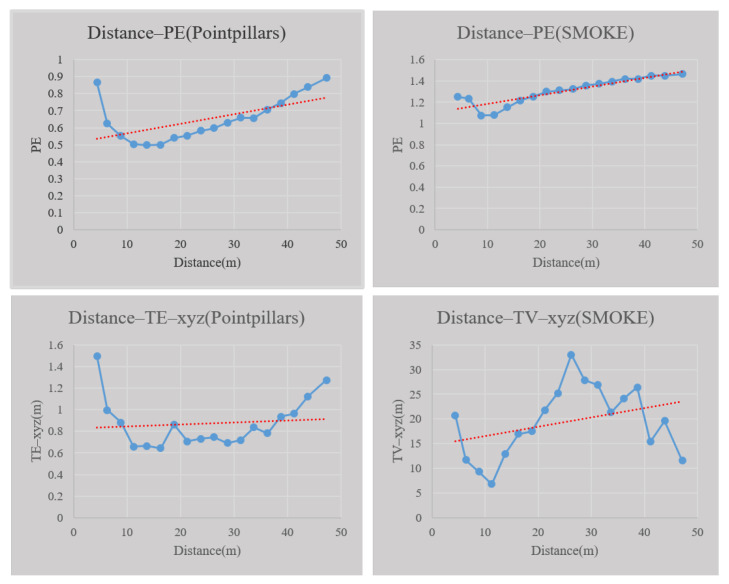
The relationship between object distance and perception uncertainty.

**Figure 16 sensors-23-02867-f016:**
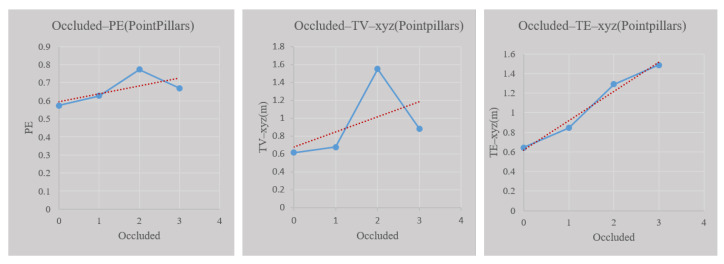
The relationship between object occlusion and perception uncertainty.

**Table 1 sensors-23-02867-t001:** Statistical mean index after matching the ground truth and perception results of PointPillars and SMOKE after DE (3769 frames).

Net	Precision	Recall	F1-Score
PointPillars	86.8%	57.8%	0.6939
SMOKE	76.5%	47.1%	0.583

**Table 2 sensors-23-02867-t002:** The number of detected networks and the average object score of PointPillars and SMOKE after DE (3769 frames).

Net	NumNET	MeanScore
PointPillars	4.375	0.6748
SMOKE	2.8659	0.4388

**Table 3 sensors-23-02867-t003:** Perception effectiveness judgment results (1000 frames).

Correct Judgment	Wrong Judgment	Failure Diagnosis Rate
920 frames	80 frames	92%

## Data Availability

KITTI dataset: https://www.cvlibs.net/datasets/kitti/ (accessed on 16 March 2022).

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
