# Peer review of "Real-Time Evaluation of Perception Uncertainty and Validity Verification of Autonomous Driving"

_sensors, 2023, doi:10.3390/s23052867_

Round 1

Reviewer 1 Report

The manuscript covers the topic of real-time evaluation of perception uncertainty and validity verification in the context of autonomous driving.
The paper is generally written well, some typos and grammar mistakes are present, which should be resolved by a spell/grammar check. References are missing at some points. The introduction introduces the reader into the topic of the study in an appropriate way and the related work section covers the most important works. There are some specific aspects that should be considered:

- if you generalize that only three modules are included in autonomous driving you should give a reference
- DNN algorithms have better perception performance compared to what?
- reference or more detailed explanation for why randomization-based approaches are better suited for distribution and parallel computation should be given
- references for objects matching algorithms should be given (Multi source perception mutual acceptance, multi-source perception mutual verification)
- rationale behind detection range of 50m and for the coordinates of the object points (500,500) and (499,499)
- more information about the training of the networks should be given
- it would be interesting to include an early-fusion network in the evaluation

The results presented in the results section are not that easy to follow and should be discussed more. In the conclusion, a a more critical view on the investigation should be provided.

Author Response

Dear Editor, and Reviewers,

We would like to thanks for your constructive and valuable comments concerning our manuscript. After carefully checking our email and the attachments, we received comments from the Editors and  Reviewers. We have studied the comments and rechecked our paper according to your suggestions and comments. The flaws have been revised one by one.

Reviewer 2 Report

The paper proposes a real-time object recognition technology in an autonomous driving system.

The author proposes multi-source perception and deep ensembal to increase the success rate of real-time object recognition.

In addition, authors are verifying its performance using the KITTI dataset, a certified dataset.

It is well structured with a description of the system proposed in the paper and its elements.

In addition, the explanation of the algorithm proposed by the author and the analysis of the results are well done, so I think it is okay to accept it as the corresponding version.

Author Response

(The authors gave the same response as above.)

Reviewer 3 Report

1- An abstract should summarize your project, methods, findings, and conclusions for the reader. The abstract of this work, however, lacks strength and is unintelligible. I recommend revising the abstract to reflect the following ideas: The overarching goal of the article and the research issues you looked at should be brief. The study's fundamental layout. Significant discoveries or trends made as a result of the research. finally, a succinct breakdown of your analyses and findings.

4- The introduction is uninteresting, disjointed, and confusing. It's essential to concentrate on the introduction, convey the events in order, and be readable. Long sentences should not be used since the intended meaning is lost. Consequently, I recommend rewriting the introduction professionally while taking the following to answer the following questions:

Q1: How can you evaluate the presented results according to other studies?’ Prove that the literature review lacks such a study by more modern references.

Q2: ‘What is the importance of the presented paper?’

Q3: What is the main challenge and issues in this study?

Q4: What is the criticism and gap analysis for academic literature that attempts to provide a solution?’

Q5: What are the recommended solutions for such challenges and their issues?’

Q6: What are the present study's implications, contributions, and novelty?’

5- The "Overview of the Proposed Methodology" section is uninteresting, disjointed, and confusing. It's essential to concentrate on the introduction, convey the events in order, and be readable. Rewrite this section and try to list (numbering) the steps to be clearer. Use a flow chart to explain the proposed approach.

6- rewrite the conclusion and consider the following comments:

-   Highlight your analysis and reflect only the important points for the whole paper.

-   Mention the benefits.

-   Mention the implication at the last of this section.

7- Figure 2, and 6 are not clear. Also, the dark blue blocks is not suitable for all figure

8- The paper contains errors and typos. Remove the replicated sentences from the whole article and correct typos.

9- The English level is low. I suggest proofreading by a specialist agent.

10- Try adding other keywords.

This paper is interesting and valuable, but some minor revisions may be necessary. Please carefully revise my comments.

Author Response

(The authors gave the same response as above.)

Round 2

Reviewer 1 Report

The authors considered the reviewer's comments appropriately and modified the manuscript accordingly. There is only one typo I like to highlight: two times it is written "Geforec", which has to be changed to "GeForce"